# An Image Encryption Transmission Scheme Based on a Polynomial Chaotic Map

**DOI:** 10.3390/e25071005

**Published:** 2023-06-29

**Authors:** Yanpeng Zhang, Wenjie Dong, Jing Zhang, Qun Ding

**Affiliations:** 1Electronic Engineering College, Heilongjiang University, Harbin 150080, China; 2211790@s.hlju.edu.cn (Y.Z.);; 2School of Information Engineering, Suihua University, Suihua 152000, China; 3Beijing Aerospace Institute of Automatic Control, Beijing 100854, China

**Keywords:** Lyapunov exponent, dynamic degradation, image encryption

## Abstract

Most existing chaotic systems have many drawbacks in engineering applications, such as the discontinuous range of chaotic parameters, weak chaotic properties, uneven chaotic sequence outputs, and dynamic degradation. Therefore, based on the above, this paper proposes a new method for the design of a three-dimensional chaotic map. One can obtain the desired number of positive Lyapunov exponents, and can also obtain the desired value of positive Lyapunov exponents. Simulation results show that the proposed system has complex chaotic behavior and high complexity. Finally, the method is implemented into an image encryption transmission scheme and experimental results show that the proposed image encryption scheme can resist brute force attacks, correlation attacks, and differential attacks, so it has a higher security.

## 1. Introduction

Numerous issues prevent the use of chaotic systems in engineering [1,2,3,4,5,6]. Firstly, since classical chaotic systems have a relatively basic structure, it is possible to predict the chaotic system’s control parameters and even the complete sequence as computing power improves. Secondly, these chaotic systems all exhibit weak chaotic phenomena, where small changes in control parameters can lead to the disappearance of chaotic phenomena. Thirdly, due to the issue of dynamic degradation, these chaotic systems pose serious security vulnerabilities, which can affect chaotic-based applications.

The application of chaos in cryptography has become a hot research field owing to its unpredictability, sensitivity to initial values, inherent randomness, ergodicity, topological transitivity, positive Lyapunov exponent, and other good properties [1,2,7,8,9]. Numerous novel chaotic systems have been developed in an effort to enhance the dynamic properties. Two general categories may be used to describe the new chaotic systems. The first category involves making suggestions for enhanced chaotic systems based on current chaotic systems. In reference [10], for the creation of a new two-dimensional chaotic map, a discrete memory model is coupled with a one-dimensional chaotic map. References [11,12] proposed nesting three different one-dimensional maps to generate segmented functions. References [13,14] coupled two one-dimensional chaos maps to generate two-dimensional chaotic maps, etc. The second type is to construct new chaotic maps from linear function or nonlinear functions. A novel two-dimensional polynomial chaotic map constructed according to “periodic three implication chaos” was proposed in reference [15], and subsequently studied the dynamic characteristics of this map. Moreover, reference [16] constructed a chaotic transmission scheme based on a three-dimensional hyperchaotic system. Reference [17] proposed an image encryption based on a three-dimensional piecewise map. However, these system models cannot obtain the desired Lyapunov exponent; that is, the Lyapunov exponent is independent of the control parameters of the system. Next, a multidimensional polynomial chaos system based on the similarity matrix is proposed in reference [18]. However, the elements of the Jacobian matrix of the system are all constant; that is, the expression of the map is linear. This method can only be applied to high-dimensional chaotic maps, but not to two-dimensional chaotic maps. In addition, if you want a different Lyapunov exponent, you have to calculate it all over again. Therefore, there is no generality. Hua et al. [19] proposed a generalized two-dimensional polynomial chaotic map, and based on this construction method, one can obtain a series of two-dimensional chaotic maps with any desired positive Lyapunov exponents. However, the last term of the constructed system is a linear term, so if the initial value and control parameters are improperly selected, then the value of the last term will collapse to a fixed value, and then the dynamic characteristics of the system will degenerate. Hua et al. [20] carried out modular transformation based on two-dimensional Henon map. After the improvement, the Lyapunov exponents of the system was directly determined by the parameter b; when the parameter b is increased, the complexity of the system was further improved. In addition, this reference proposed the optimization of the Henon map model; that is, the optimization of the existing model. In turn, the application is more limited. In order to solve the above problems, a generalized three-dimensional polynomial chaotic system is proposed in this paper.

The following is a summary of the main work of this paper: (1) We propose a generalized three-dimensional polynomial chaotic map model. By giving different control parameters and the highest degree of polynomial through a proposition, a series of three-dimensional chaotic maps can be obtained. Moreover, these systems are robust and can obtain the expected Lyapunov exponent. (2) The theoretical analysis results indicate that the three-dimensional polynomial chaotic map model can generate robust chaos and the expected Lyapunov exponent. (3) The transmission of encryption and decryption images is realized by synchronization of chaotic systems.

The remaining parts of this article are organized as follows. A three-dimensional polynomial chaotic map model is shown in Section 2, and the parameter range for the existence of chaotic behavior is provided by a proposition. A transmission mechanism for image encryption is provided in Section 3. Section 4 provides the related security analysis. The conclusion is presented in Section 5.

## 2. Construction of 3D Polynomial Chaotic System Model

A new three-dimensional polynomial dynamical map is designed in this section to improve the dynamic properties of the system. One can obtain the desired number of positive Lyapunov exponents, and can also obtain the desired value of positive Lyapunov exponents. The mathematical equations of a three-dimensional polynomial dynamical map proposed in this paper are calculated as follows:(1)$$\{\begin{array}{l} x_{n+1}=\left(ax_{n}+by_{n}{}^{\gamma }\right)\mathrm{mod}\beta \\ y_{n+1}=\left(cy_{n}+dz_{n}^{\gamma }\right)\mathrm{mod}\beta \\ z_{n+1}=\left(ez_{n}+r\right)\mathrm{mod}\beta \end{array},$$
where $x_{n},y_{n}$, and $z_{n}$ are the state variables of the system (1), and the control parameters of the proposed system are a,b,c,d,e,r; moreover, γ is the highest degree of the polynomial; β is the modulus coefficient. In this paper, r is taken as a random disturbance to prevent state variable $z_{n}$ from collapsing into a fixed value. Next, we discuss the range of chaotic parameters of the dynamical system with Proposition 1. The most reliable way to determine whether a map is chaotic or not is to use the Lyapunov exponent; therefore, the map proposed in this paper was evaluated by using the Lyapunov exponent.

**Definition** **1****([21]).** *A three-dimensional map is considered chaotic if it is globally bounded, and has at least one positive Lyapunov exponent; if there is more than one positive Lyapunov exponent, it is hyperchaotic*.

The three Lyapunov exponents of a three-dimensional discrete dynamic system are calculated as follows [22]:(2)$$LE_{i}=\underset{k\to \infty }{\lim}\frac{1}{k}\ln\lambda_{i}\left(\Phi_{k}\right),i=1,2,3$$
where $\lambda_{i}\left(\Phi_{k}\right)$ is the *i*-th eigenvalue of matrix $\Phi_{k}$, and the expression for $\Phi_{k}$ is as follows:(3)$$\Phi_{k}=\prod_{i=0}^{k-1}J\left(x(i),y(i),z\left(i\right)\right),$$
where J(·) is the Jacobian matrix at the *i*-th iteration of the system.

**Proposition** **1.**
*If any of the three control parameters of system (1) are satisfied such that |a|>1,|c|>1 and |e|>1, then the system is chaotic; if any two of the three control parameters of system (1) are satisfied such that |a|>1,|c|>1 and 
|e|>1, the system is hyperchaotic.*


**Proof.** The Jacobian matrix of the *i*-th iteration of system (1) is as follows:
(4)$$J\left(x(i),y(i),z\left(i\right)\right)=\left(\begin{array}{ccc} a & b\cdot \gamma \cdot y_{i}^{\gamma -1} & 0\\ 0 & c & d\cdot \gamma \cdot z_{i}^{\gamma -1}\\ 0 & 0 & e \end{array}\right)$$
and the expression for matrix $\Phi_{k}$ is derived as follows:(5)$$\begin{array}{ll} \Phi_{k} & =\prod_{i=0}^{k-1}J\left(x(i),y(i),z\left(i\right)\right)=\left(\begin{array}{ccc} a & b\cdot \gamma \cdot y_{0}^{\gamma -1} & 0\\ 0 & c & d\cdot \gamma \cdot z_{0}^{\gamma -1}\\ 0 & 0 & e \end{array}\right)\times \cdots \times \left(\begin{array}{ccc} a & b\cdot \gamma \cdot y_{k-1}^{\gamma -1} & 0\\ 0 & c & d\cdot \gamma \cdot z_{k-1}^{\gamma -1}\\ 0 & 0 & e \end{array}\right)\\ & =\left(\begin{array}{ccc} a^{k} & t_{1} & t_{2}\\ 0 & c^{k} & t_{3}\\ 0 & 0 & e^{k} \end{array}\right) \end{array}$$
where $t_{1},t_{2},$ and $t_{3}$ are specific numbers expressed by the variables and control parameters. Then the two eigenvalues $\lambda_{1},\lambda_{2},$ and $\lambda_{3}$ of the matrix $\Phi_{k}$ can also be easily solved.
(6)$$\lambda_{1}=a^{k},\lambda_{2}=c^{k},\lambda_{3}=e^{k}$$
Then, the three Lyapunov exponents of the map are derived as follows:(7)$$LE_{1}=\underset{k\to \infty }{\lim}\frac{1}{k}\ln\left(\lambda_{1}\right)=\underset{k\to \infty }{\lim}\frac{1}{k}\ln{\left(a\right)}^{k}=\ln\left(a\right).$$
By the same token, $LE_{2}=\ln\left(c\right),LE_{3}=\ln\left(e\right)$. The output of the system must be globally bounded due to the modulo operation in the model. If control parameter |a|>1, then the map is chaotic. If control parameter |a|>1 and |e|>1, the map is hyperchaotic. □

### 2.1. Numerical Example

In order to facilitate calculation, we take γ=2 in this paper, and the mathematical expression of the three-dimensional polynomial chaotic map is as follows:(8)$$\{\begin{array}{c} x_{n+1}=\left(ax_{n}+by_{n}^{2}\right)\mathrm{mod}\beta \\ \begin{array}{l} y_{n+1}=\left(cy_{n}+dz_{n}^{2}\right)\mathrm{mod}\beta \\ z_{n+1}=\left(dy_{n}+r\right)\mathrm{mod}\beta \end{array} \end{array}.$$

According to Proposition 1, if two or more of |a|>1,|c|>1, and |e|>1 are true, the system is in hyperchaos; if one of them is true, the system is in a chaotic state; if none of them is true, the system is in a stable state. Figure 1 shows the trajectory of the proposed map with following parameters: a=2,b=−0.2,c=3,d=1.2,e=1.7,β=1, and $r_{}$ is a random number between (0,1). With the increase of iteration times, the output of the map can randomly visit or approach all the areas of the data range. The bifurcation diagram of a dynamic system shows the points that the system passes through under different parameters, and this process provides an intuitive way for scholars to study the nature of chaos. Figure 2 illustrates in three-dimensional terms the bifurcation diagrams for different parameters, and it can be seen that the state variable $x_{n}$ is evenly distributed throughout the space for different control parameters. The Lyapunov exponents of the proposed map varying with the control parameters a and c are shown in Figure 3. One can observe that the proposed map has three positive Lyapunov exponents in the parameter range. In addition, if the control parameters of the map change slightly, the proposed map is still in chaos, which can indicate that the proposed map shows robust hyperchaotic behavior and more complex dynamic properties.

### 2.2. Sample Entropy Analysis

The complexity of dynamical system refers to the degree to which the time sequence is close to the random sequence. The higher the complexity, the closer the sequence is to the random sequence and the higher the corresponding security. In this paper, sample entropy is adopted to calculate the degree of complexity of a time series [23,24]. In order to give a better description, the sample entropy was normalized later. The closer the sample entropy is to 1, the greater the irregularity of the map is. Figure 4 shows the sample entropy values of the three sequences of the map as the parameters change a and c. It is not difficult to see that the sample entropy values in Figure 4 are close to 1 in the whole interval, indicating that the chaotic sequence generated by the map has high complexity and can be applied to the fields of image encryption, information processing, and secure communication.

### 2.3. Pseudo-Randomness Analysis

In this section, the NIST SP 800-22 tests are used to demonstrate the randomness of the output sequences of the proposed maps. The 15 tests include frequency, longest run, approximate entropy test, linear complexity test, and so on. Additionally, these 15 tests focus on whether the binary sequence has an acceptable pseudo-randomness [25]. The binary sequence is considered random if the estimated *p*-value is 0.01, otherwise it is considered non-random. As the output of the proposed map is in the range [0, 1], it is recommended that a pseudo-random number generator be used, which has the following structure:(9)$$B_{i}=\left\lfloor d_{i}\times \alpha \right\rfloor \mathrm{mod}\beta ,i=1,2,3$$
where ⌊x⌋ is used to obtain the largest integer smaller than x or equal to x, and $d_{i}$ is the three sequences of the proposed map. In addition, α is a large number that affects the value of the sequence, and β is an integer. Set $\alpha =10^{8},\;\beta =256,$ and we can obtain a binary sequence of 8 bits.

After the system (9) has been quantized by the pseudo-random sequence generator introduced in this paper, there are three pseudo-random sequences generated, for which the first 3000 values are dropped to avoid the initial value effect. Then, 100 sets of sequences of length $10^{6}$ are taken and tested with the NIST test suite, respectively. Table 1 shows the set with the lowest *p*-values. Therefore, it is clearly shown in the Table 1 that all 15 tests were successful. This shows that the chaotic sequence generated by this model has strong randomness, which indicates that the sequence produced by this model is random relative to the 15 tests of the NIST suite.

## 3. A 3D Polynomial Chaotic Image Encryption Transmission Scheme

### 3.1. Image Encryption Scheme

The proposed map-based image encryption model is presented in this section. The encryption scheme is based on “confusion” and “diffusion”, with the confusion part working by separating adjacent pixels in the image to different positions and the diffusion part using an invertible function to change the value of a specific pixel. This paper performed the aforementioned process twice. The confusion part is described in detail by Algorithm 1 below. A numerical example is presented in Figure 5. Matrices $X_{L}$ and $X_{L}$ are reshaped by the chaotic sequences X and Y, whose lengths are $4^{2}.$ It can be observed that almost all pixels are scrambled after a round of confusion. The diffusion part will process the confusion image F again. In one round of encryption, one can rearrange the confusion image F into one column matrix $F_{1D}$, and sort the $F_{1D}$ with $I_{2}$ as a matrix A. Figure 6 shows a numerical example of the scheme, and the current pixel of the diffusion image can be obtained by:(10)$$D_{i}=\{\begin{array}{cc} \left\lfloor \left(A_{i}+A_{L\times L}+\left|Y_{i}\right|\times 2^{32}\right)\mathrm{mod}256\right\rfloor & \;\mathrm{if}\;i=1,\\ \left\lfloor \left(A_{i}+D_{i-1}+\left|Y_{i}\right|\times 2^{32}\right)\mathrm{mod}256\right\rfloor & \mathrm{if}\;i\in \left[2,L\times L\right], \end{array}$$
where ⌊x⌋ is used to obtain the largest integer that is smaller than x or equal to x. In addition, one can rearrange D into a matrix with size of L×L. The second round of operations performs the same operation based on another matrix, and the encrypted ciphertext image can be obtained.
**Algorithm 1:** The procedure of the confusion part of the proposed image encryption scheme.**Input:** Plaintext image P and initial values x(0), y(0), z(0).**Output:** Confusion image F.Truncate the output sequence of the proposed map as size $L^{2}$, where L×L is the size of the image.Reshape the sequences
X, Y, and Z in columns into L×L matrices, denoted as $X_{L}$, $Y_{L}$, and $Z_{L}.$Matrices $S_{1}=X_{L}\times Y_{L}$ and $S_{2}=Z_{L}$ can be obtained.Sort $S_{1}$ and $S_{2}$ in ascending order, and obtain their index vectors $I_{1}$ and $I_{2}.$Rearrange the pixel locations of plaintext image P by using the index matrix above.The confusion image F is obtained.

### 3.2. Image Decryption Scheme

Generally speaking, the decryption process is the inverse operation of the encryption process. Thus, the process of diffusion can be described as follows:(11)$$F_{i}=\{\begin{array}{cc} \left\lfloor \left(D_{i}-D_{i-1}-\left|Y_{i}\right|\times 2^{32}\right)\mathrm{mod}256\right\rfloor & \mathrm{if}i\in \left[2,L\times L\right],\\ \left\lfloor \left(D_{i}-A_{L\times L}-\left|Y_{i}\right|\times 2^{32}\right)\mathrm{mod}256\right\rfloor & \mathrm{if}i=1. \end{array}$$
Thus, the confusion image can be obtained using the inverse operation of diffusion of image encryption. What is more, the original image can be completely reconstructed using the inverse confusion of image encryption.

### 3.3. Nonlinear Feedback Synchronization Control Scheme

A polynomial discrete chaotic system is taken as an example to realize the synchronization of the nonlinear feedback method. The driving system is:(12)$$\{\begin{array}{c} x_{1}\left(n+1\right)=\left(2x_{1}\left(n\right)-0.2x_{2}^{2}\left(n\right)\right)\mathrm{mod}\beta \\ \begin{array}{l} x_{2}\left(n+1\right)=\left(3x_{2}\left(n\right)+1.2x_{3}^{2}\left(n\right)\right)\mathrm{mod}\beta \\ x_{3}\left(n+1\right)=\left(1.7x_{3}\left(n\right)+r\right)\mathrm{mod}\beta \end{array} \end{array},$$
and the expression of the response system is:(13)$$\{\begin{array}{c} y_{1}\left(n+1\right)=\left(2y_{1}\left(n\right)-0.2y_{2}^{2}\left(n\right)+u_{1}\left(t\right)\right)\mathrm{mod}\beta \\ \begin{array}{l} y_{2}\left(n+1\right)=\left(3y_{2}\left(n\right)+1.2y_{3}^{2}\left(n\right)+u_{2}\left(t\right)\right)\mathrm{mod}\beta \\ y_{3}\left(n+1\right)=\left(1.7y_{3}\left(n\right)+r+u_{3}\left(t\right)\right)\mathrm{mod}\beta \end{array} \end{array},$$
where ${\left[u_{1},u_{2},u_{3}\right]}^{T}$ is a vector controller. We aim to design a suitable nonlinear controller so that the state trajectory of the slave system is consistent with that of the master system, namely $\lim_{n\to \infty }\left\|y_{n}-x_{n}\right\|=0$. The vector controller is designed as follows:(14)$$\{\begin{array}{l} u_{1}=-0.5\left(2y_{1}\left(n\right)-2x_{1}\left(n\right)\right)-\left(-0.2y_{2}^{2}\left(n\right)+0.2x_{2}^{2}\left(n\right)\right)\\ \begin{array}{l} u_{2}=-0.7\left(3y_{2}\left(n\right)-3x_{2}\left(n\right)\right)-\left(1.2y_{3}^{2}\left(n\right)-1.2y_{3}^{2}\left(n\right)\right)\\ u_{3}=-0.9\left(1.7y_{2}\left(n\right)-1.7x_{2}\left(n\right)\right) \end{array} \end{array}$$

Defining systematic errors: $e_{i}(n)=y_{i}(n)-x_{i}\left(n\right),\left(i=1,2,3\right)$, and the synchronous error discrete system of the drive system (12) and the response system (13) is expressed as Equation (15).
(15)$$\{\begin{array}{c} e_{1}\left(n+1\right)=0.5e_{1}(n)\\ \begin{array}{l} e_{2}\left(n+1\right)=0.3e_{2}(n)\\ e_{3}\left(n+1\right)=0.1e_{2}(n) \end{array} \end{array}$$

**Lemma** **1****([26]).** *For linear discrete difference systems* e(n)=Ae(n−1), A∈ℝn×n *is a coefficient matrix. The system is asymptotically stable if the magnitude of all eigenvalues of matrix* A *a is less than or equal to 1.*

Obviously, the Equation (15) is asymptotically stable; that is, system (12) and system (13) can achieve synchronization. We set the initial values of systems (12) and (13) as $\left(x_{1},x_{2},x_{3}\right)=(0.4,0.6,0.5)$ and $\left(y_{0},y_{2},y_{3}\right)=(0.3,0.2,0.4)$, respectively. The performance of the synchronization error is shown in Figure 7. The synchronization error is approaching 0 with a quickly speed, in consequence the transceiver system achieves synchronization.

### 3.4. Transmission Scheme

An image encryption transmission based on nonlinear feedback synchronization is proposed in this paper. The framework of the proposed scheme is depicted in Figure 8, in which the proposed three-dimensional polynomial chaotic system (12) is used as the master system and the slave system is system (13). For a gray image, one can rearrange the encrypted image D into one column matrix $D_{1D}$. Then, the sequence $D_{1D}$ and state variable $x_{1}\left(i\right)$ are masked into the signal $E_{n}$; then, the signal $E_{n}$ is sent to the receiver end via the public channel. At the receiver end, the recovered signal $D_{1D}^{\prime }$ can be obtained through the signal $E_{n}$ and state variable $y_{1}\left(i\right)$. In addition, there will inevitably be noise in the channel to break the transmitted signal. Taking Gaussian white noise as an example, we add a DCT transform filter to the receiving end to ensure that the recovered image is as close as possible to the original image. Finally, one can also reshape the recovered signal $D_{1D}^{\prime }$ into an image with size of L×L.
(16)$$E_{n}=D_{1D}+x_{1}\left(i\right)\times 10^{5},$$
(17)$$D_{1D}^{\prime }=E_{n}-y_{1}\left(i\right)\times 10^{5}.$$

### 3.5. Simulation Results

Different kinds of encrypted pictures must be transmitted to the receiving side over a public channel in the image encryption transmission system. The receiver must also evenly recreate the original picture. Various kinds of pictures encrypted using the suggested approach are displayed in Figure 9.

## 4. Security Analysis

There are a number of analyses, such as key security analysis, histogram analysis, the Shannon entropy analysis, correlation analysis, and differential attacks, that can be applied to show the performance of the proposed image encryption scheme. Therefore, this section applies the above methods to measure the performance of the introduced methods.

### 4.1. Key Sensitivity Analysis

An image encryption scheme should firstly have a large enough key space to resist brute-force attack. The security key of the proposed encryption scheme is a binary string with 320 bits. The security key contains 10 parts $\left\{a,b,c,d,e,x_{0}+h,y_{0}+h,z_{0}+h,r,\beta \right\}$, where h is the hash value (SHA-256) of the plaintext information. Since their lengths are 32 bits, respectively, and the key space of the proposed scheme can reach $2^{320}$ since the key length is 320 bits. The key space is much larger than $2^{100}$; that is, the scheme is resistant to brute-force attacks. An incorrect key with a little change from the initial key might also obtain the plaintext information, hence the suggested picture encryption system must be extremely sensitive to the initial key. In other words, each key may decode the encrypted picture when the identical plaintext image is encrypted with two keys that differ by one bit. Hence, an effective encryption technique should be capable of preventing the recovery of the original picture data during decryption using a different key. Figure 10 shows the key sensitivity results. The same plaintext image is encrypted and decrypted by two keys and with one bit difference. Each key can decrypt the original image. If the other key is used for decryption, the original image information cannot be obtained. Thus, the proposed scheme is sensitive to its keys in both encryption and decryption processes.

### 4.2. Histogram Analysis

For further evaluation of the uniformity of the pixel values of the encrypted images, in this paper we employed the chi-square test. The statistics value $\chi^{2}$ can be defined as
(18)$$\chi^{2}=\sum_{i=0}^{255}\left(\frac{E_{i}-Z}{Z}\right),$$
where $E_{i}$ is the value of the current pixel, and Z is the expected occurrence frequencies of each pixel. When the calculated $\chi^{2}$ value of a ciphertext image does not exceed 293.2478, the encrypted image can pass the chi-square assessment [27]. The chi-square values of virous encrypted images are shown in Table 2. Obviously, those values do not exceed 293.2478, which shows that the distributions of the pixel values of the encrypted images are uniformly distributed.

### 4.3. The Shannon Entropy

In order to quantitatively measure the information distribution of the ciphertexts, the Shannon entropy is applied to assess whether an encrypted image is a random-like image with pixel values randomly distributed. Its mathematical expression is defined by:(19)$$H\left(R\right)=-\sum_{i=0}^{F-1}P\left(R=i\right)\log_{2}P\left(R=i\right),$$
where F and R are the maximum and individual pixel values of an image, respectively. In addition, P(⋅) is the discrete probability density function. Take gray images, for example, F=256 and each pixel contains 8 binary bits: when P(R=i)=1256, and H(R)=8; that is, the encrypted image is uniformly distributed. Table 3 lists the Shannon entropy of several test images obtained from the USC-SIPI image database. It can be seen from Table 3 that the average Shannon entropy value of the image after encryption by this scheme is 7.9986, and these values are much closer to 8, which indicates the encrypted images are uniformly distributed.

### 4.4. Correlation Analysis

The pixel correlation of an image covers three directions: horizontal, vertical, and diagonal. Thus, for a good encryption algorithm, the goal should be to reduce the correlation between adjacent pixels. It can be defined as the correlation between two pixel sequences, which is given by:(20)$$r_{uv}=\frac{\mathrm{cov}\left(u,v\right)}{\sqrt{D\left(u\right)}\sqrt{D\left(v\right)}},$$
(21)$$\mathrm{cov}\left(u,v\right)=\frac{1}{N}\sum_{i=1}^{N}\left(u_{i}-E\left(u\right)\right)\left(v_{i}-E\left(v\right)\right),$$
(22)$$D\left(u\right)=\frac{1}{N}\sum_{i=1}^{N}{\left(u_{i}-E\left(u\right)\right)}^{2},$$
(23)$$E\left(u\right)=\frac{1}{N}\sum_{i=1}^{N}u_{i},$$
where u and v are adjacent pixels values, $r_{uv}$ is correlation coefficient of the adjacent pixels. The 3000 pairs of adjacent pixels—from the plaintext and encrypted images in horizontal, vertical, and diagonal directions—are randomly selected. These distribution of the 3000 pairs are shown in Figure 11. The plaintext image has pixels close to the diagonal while the cipher text image has a random distribution of pixels which can be seen in Figure 10. The comparative results of the correlation obtained by using different encryption schemes are presented in Table 4. It is clear that the proposed method has an $r_{uv}$ value close to zero compared to the other schemes.

### 4.5. Differential Attack

When little modifications to the source picture cause substantial changes to the encrypted image, differential attacks are largely ineffective. The number-of-pixels change rate (NPCR) and unitary averaged changed intensity (UACI) tests [28] are used to assess the capacity of the proposed picture encryption methods to withstand differential assaults. The NPCR and UACI can be expressed as shown below:(24)$$NPCR=\frac{\sum_{m=1}^{M}\sum_{n=1}^{N}D\left(m,n\right)}{MN}\times 100\%,$$
(25)$$D\left(m,n\right)=\{\begin{array}{cc} 1, & \mathrm{for}C_{1}\left(m,n\right)\neq C_{2}\left(m,n\right);\\ 0, & \mathrm{otherwise}. \end{array}$$
(26)$$UACI\left(C_{1},C_{2}\right)=\sum_{m=1}^{M}\sum_{n=1}^{N}\frac{\left|C_{1}\left(m,n\right)-C_{2}\left(m,n\right)\right|}{255\times M\times N}\times 100\%,$$
where denotes the two encrypted pictures $C_{1}$ and $C_{2}$, which are identical to the original images except for a single additional pixel, and D(m,n) is the total number of pixels in the encrypted images $C_{1}$ and $C_{2}.$ The ideal expectations NPCR and UACI values are 99.61% and 33.46%, respectively [12]. The suggested encryption scheme’s mean NPCR and UACI values are shown in Table 5, and the size of all of the pictures is 512×512, from the USC-SIPI’s Miscellaneous dataset, along with the comparisons to other methods. The findings are shown in Table 6. These findings are obviously more in line with the intended predicted values, which shows that the suggested method performs better in terms of defending against differential assaults.

**Table 4 entropy-25-01005-t004:** Adjacent pixel correlations of the plaintext image “Lena” and its ciphertext image using different encryption schemes.

Schemes	Horizontal	Vertical	Diagonal
“Lena” image	0.94010	0.97689	0.95667
Ref. [23]	0.00030	0.00140	0.00220
Ref. [28]	−0.00150	−0.00210	0.00190
Ref. [29]	0.00283	0.00183	0.00330
Ref. [30]	0.00340	0.00580	0.00450
Ref. [31]	−0.00150	0.00410	0.00690
Proposed method	−0.00091	−0.00110	0.00100

**Table 5 entropy-25-01005-t005:** The values of NPCR and UACI of ciphered images.

Images	NPCR (%)			UACI (%)		
	R	G	B	R	G	B
4.1.01.tiff	99.60	99.61	99.62	33.14	33.17	33.47
4.1.03.tiff	99.61	99.62	99.63	33.34	33.67	33.42
4.1.04.tiff	99.64	99.60	99.64	33.25	33.36	33.43
4.2.03.tiff	99.62	99.59	99.62	33.15	33.46	33.42
4.2.07.tiff	99.58	99.61	99.61	33.24	33.53	33.43
Lena	99.60	99.58	99.63	33.41	33.32	33.45

**Table 6 entropy-25-01005-t006:** NPCR and UACI comparison of Lena.

Reference	NPCR (%)			UACI (%)		
	R	G	B	R	G	B
Ref. [32]	99.62	99.60	99.64	33.50	33.47	33.43
Ref. [33]	Mean = 99.64			Mean = 33.49		
Ref. [34]	99.63	99.68	99.69	33.45	33.42	33.46
Ref. [35]	Mean = 99.63			Mean = 33.47		
This paper	99.67	99.64	99.67	33.42	33.43	33.44

### 4.6. Complexity Analysis

Because the system is a discrete time model, that is, a difference equation, the time complexity of this part is O(n), where $n=L^{2}$. The confusing part is the indexed sequential lookup problem, which has a complexity of O(nlogn). The diffusion part is a cycle of n degrees, and the complexity of this part is also O(n). In addition, the complexity of the hash function is O(1). Then, the time complexity of the scheme is O(nlogn). The effectiveness of the encryption technique is also significantly measured by the time complexity, based on the AMD Ryzen 7 5800 H 3.20 GHz CPU model, and the modeling environment is Matlab 2021a. The encryption technique suggested in this research was used to independently encrypt and time-test the Lena pictures of sizes 128 × 128, 256 × 256, 512 × 512 and 1440 × 900. The test results are displayed in Figure 12. The approach suggested in this work also yielded the following results: for the algorithm, the encryption time is related to the size of the picture being encrypted; the larger the image, the longer the encryption time. The scheme processes the required data in two rounds, so the computational complexity is O(n).

## 5. Conclusions

This paper presents a new approach to the design of a generalized three-dimensional chaotic model, whose Lyapunov exponents can be constructed directly from system control parameters. In addition, the proposed method was used to construct three-dimensional robust chaotic maps with different Lyapunov exponents. Simulation results showed that the chaotic system has complex chaotic behavior and high complexity. Finally, the method was implemented into an image encryption transmission scheme. Experimental results showed that the original image could be recovered from the receiver, while the proposed algorithm was analyzed for security using initial secret key sensitivity tests, histogram analysis, and differential attacks, and so on, and the simulation results demonstrated the feasibility of the proposed method.

## Figures and Tables

**Figure 1 entropy-25-01005-f001:**
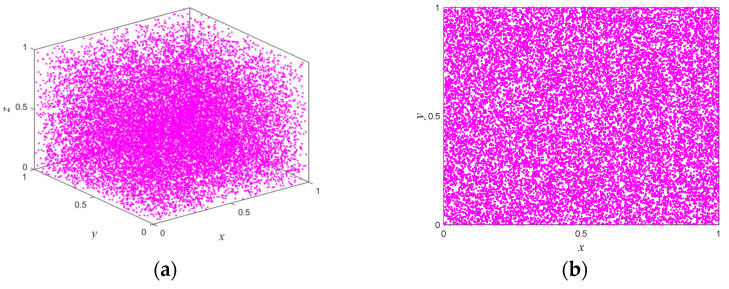
The phase trajectory of the system with initial conditions (x(0),y(0),z(0))= (0.4,0.6,0.5): (**a**) x−y−z phase diagram; (**b**) x−y phase diagram; (**c**) x−z phase diagram; (**d**) y−z phase diagram.

**Figure 2 entropy-25-01005-f002:**

The bifurcation diagram of the system with two parameters: (**a**) the bifurcation diagram of state variable $x_{n}$ with control parameters a and b; (**b**) the bifurcation diagram of state variable $x_{n}$ with control parameters a and c; (**c**) the bifurcation diagram of state variable $x_{n}$ with control parameters b and c.

**Figure 3 entropy-25-01005-f003:**
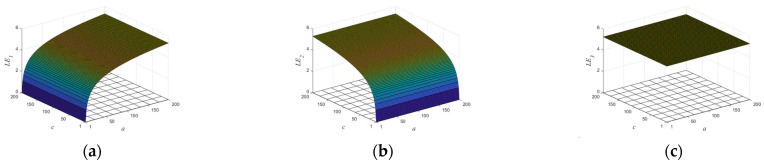
The Lyapunov exponents of the proposed map varying with the control parameters a and c with [1, 200]: (**a**) the $LE_{1}$ values of the proposed map; (**b**) the $LE_{2}$ values of the proposed map; (**c**) the $LE_{3}$ values of the proposed map.

**Figure 4 entropy-25-01005-f004:**
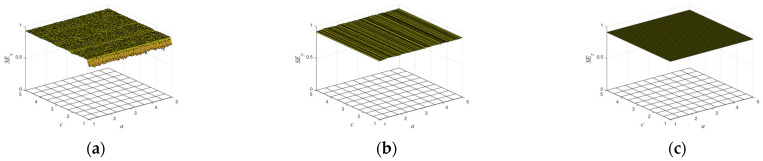
The sample entropy values of the three sequences of the map as the parameters change a and c with [1,5]: (**a**) the sample entropy values of x sequence of the map; (**b**) the sample entropy values of y sequence of the map; (**c**) the sample entropy values of z sequence of the map.

**Figure 5 entropy-25-01005-f005:**
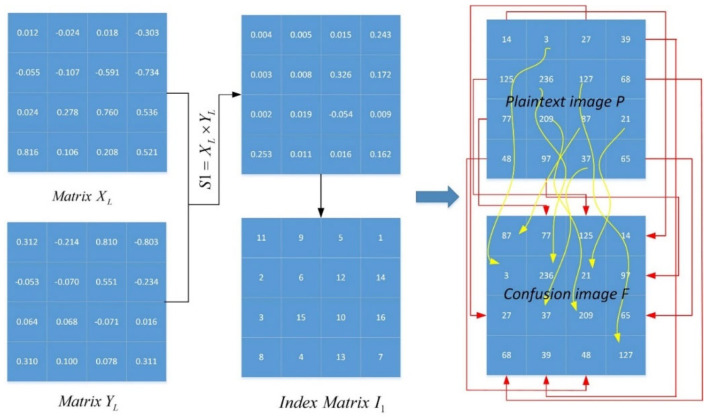
An example of the confusion part of image encryption.

**Figure 6 entropy-25-01005-f006:**
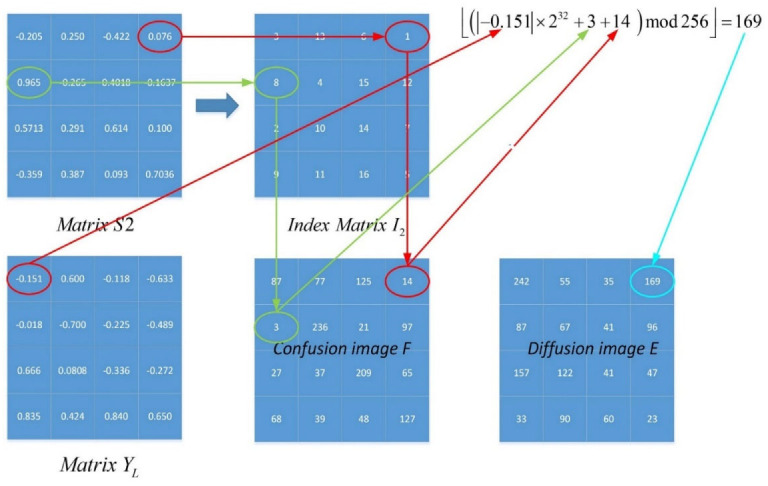
An example of the diffusion part of image encryption.

**Figure 7 entropy-25-01005-f007:**
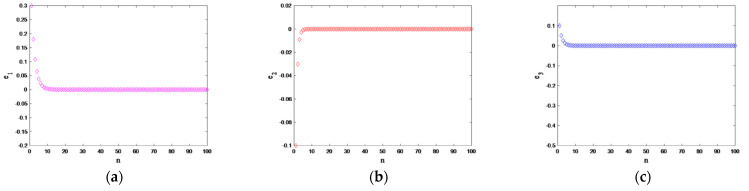
Synchronization error diagram of drive–response system: (**a**) $e_{1}$; (**b**) $e_{2}$; (**c**) $e_{3}$.

**Figure 8 entropy-25-01005-f008:**
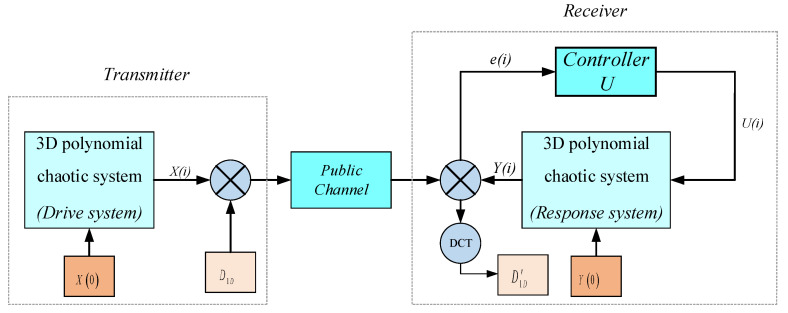
The framework of the proposed image encryption transmission scheme.

**Figure 9 entropy-25-01005-f009:**
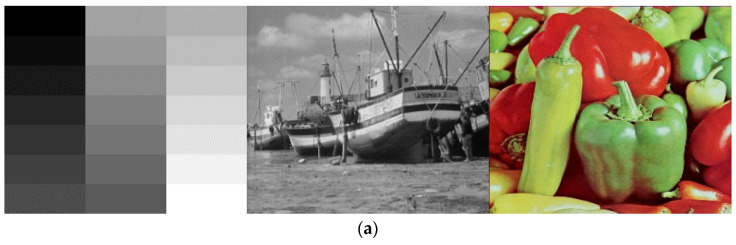
Simulation results: (**a**) plaintext images; (**b**) histograms of plaintext images; (**c**) images of the receiver end; (**d**) histogram of images of the receiver end.

**Figure 10 entropy-25-01005-f010:**
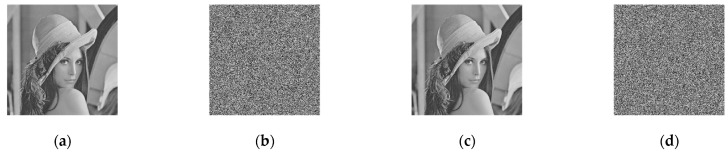
The key sensitivity results: (**a**) the plaintext image; (**b**) the ciphertext image E encrypted by $K_{1}$; (**c**) the decrypted $D_{1}$ from E using $K_{1}$; (**d**) the decrypted $D_{2}$ from $E_{1}$ using $K_{2}$.

**Figure 11 entropy-25-01005-f011:**
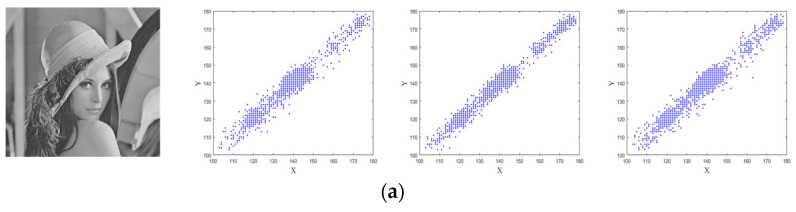
The correlation distributions: (**a**) the plaintext image and correlation distributions of three directions; (**b**) the ciphertext image and correlation distributions of three directions.

**Figure 12 entropy-25-01005-f012:**
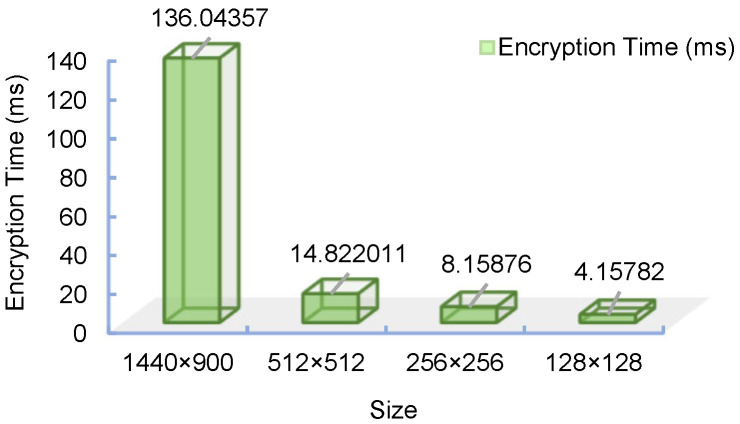
Histograms of encryption times for different image sizes.

**Table 1 entropy-25-01005-t001:** NIST test results of the proposed chaotic sequences.

	Test Suites	*p*-Value			Result
		$Seq_{x}$	$Seq_{y}$	$Seq_{y}$	
1	Frequency	0.554320	0.326810	0.457832	Pass
2	Block frequency	0.834570	0.577802	0.758221	Pass
3	Runs	0.547600	0.197506	0.421572	Pass
4	Longest run	0.801265	0.792351	0.823451	Pass
5	Rank	0.972745	0.267811	0.765341	Pass
6	FFT	0.035687	0.948721	0.689521	Pass
7	Non-overlapping template	0.235874	0.478512	0.367876	Pass
8	Overlapping template	0.497832	0.089451	0.289765	Pass
9	Universal	0.935647	0.058974	0.321768	Pass
10	Linear complexity	0.798145	0.278945	0.614729	Pass
11	Serial	0.754612	0.845971	0.792635	Pass
12	Approximate entropy	0.616784	0.089451	0.1976217	Pass
13	Cumulative sums	0.168745	0.944513	0.7122319	Pass
14	Random excursions	0.654123	0.087945	0.0933683	Pass
15	Random excursions variant	0.565209	0.058799	0.3548823	Pass

**Table 2 entropy-25-01005-t002:** The $\chi^{2}$ values of encrypted images.

Images	Lena	Gray	Ruler	Boat	Pepper
$\chi^{2}$	242.042	230.458	237.344	246.341	224.633

**Table 3 entropy-25-01005-t003:** The Shannon entropy values of several original and encrypted images.

File Name	Original Image $H_{P}$	Encrypted Image $H_{e}$
5.1.09.tiff	6.7093	7.9975
5.1.13.tiff	1.5483	7.9986
5.3.01.tiff	7.5237	7.9991
boat.512	7.1914	7.9992
ruler.512	0.5000	7.9977
gray21.512	4.3923	7.9990
Mean value	4.6442	7.9986

## Data Availability

All results and data obtained can be found in open access publications.

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
