# Peer review of "An Image Encryption Transmission Scheme Based on a Polynomial Chaotic Map"

_entropy, 2023, doi:10.3390/e25071005_

Round 1

Reviewer 1 Report

This paper designs an image encryption transmission scheme based on polynomial chaotic map, which will promote cryptographic applications of nonlinear sciences.

1. As for the issue of dynamic degradation, refer to 
https://dx.doi.org/10.1142/S0218127423500633

2. Most of references are out of date, compare the proposed scheme with the recent counterparts, e.g. https://dx.doi.org/10.1007/s00500-021-06500-y

3.  Why and how the proposed scheme can solve the security challenges of private-key encryption schemes summarized at https://doi.org/10.1016/j.jisa.2020.102566   should be explained in the introduction part.

4. The computational complexity of the proposed image encryption transmission scheme should be presented.

5. Embed every equation among a text sentence. Present every equation in a way can be found in any textbook on Calculus or famous monograph. As for the formal presentation of an equation, refer to http://linear.axler.net/LinearAbridged.pdf

NA

Reviewer 2 Report

An image encryption transmission scheme based on polyno-2 mial chaotic map was proposed in this paper, in which a generic two-dimensional chaotic model was designed. The idea is interesting with some comments as,

1, Can the 2D be extended to 3D for a large key space?

2, How about the time cost? It can be added.

3, In section of Security analysis, color images are better shown.

4, Some new methods in recent years are better described in the content, such as “Frontiers of Computer Science, 2023, 17(3): 173804.”” https://doi.org/10.1016/j.sigpro.2023.109107”. And do some comparisons.

An image encryption transmission scheme based on polyno-2 mial chaotic map was proposed in this paper, in which a generic two-dimensional chaotic model was designed. The idea is interesting with some comments as,

1, Can the 2D be extended to 3D for a large key space?

2, How about the time cost? It can be added.

3, In section of Security analysis, color images are better shown.

4, Some new methods in recent years are better described in the content, such as “Frontiers of Computer Science, 2023, 17(3): 173804.”” https://doi.org/10.1016/j.sigpro.2023.109107”. And do some comparisons.

Reviewer 3 Report

Summary:

In this paper, a new two-dimensional chaos model is proposed to solve the problems of the existing chaos system, and a method for transmitting images by encryption is presented. The proposed method provides higher encryption security, and it has been experimentally demonstrated that secure image transmission is possible. Moreover, the security of the proposed method is confirmed by analysis such as initial key sensitivity analysis, histogram analysis, correlation analysis and differential attack.

Performance: the objective is to reduce the correlation of neighboring pixel values by performing correlation analysis of the method. These analysis results show that the proposed method is a secure and effective image encryption method.

A look at the performance comparison in Table 3 shows that the method described in this paper proves to be effective.

However, it is unfortunate that there are no evaluation figures with which to compare the data in Table 4.

Reviewer 4 Report

The article deals with the topic of chaotic cryptography. The authors present a new dynamical system that is used to encrypt images.

The article is interesting, and, despite some unclear fragments, it is quite readable. However, some issues require the authors' attention:

1) In formula 9, the authors use the term integral function - what does it mean in this context? Or is it a typo and is it a floor-type function?

2) For NIST tests, please specify how many bit sequences were generated and what length they were.

3) Please explain the symbols used in formula 10. Isn't line 147 about formula 10 instead of 11?

4) What exactly does the decryption process look like? Please describe it using algorithms, as is the case with the encryption process.

5) Figure 5: The histograms for the Peppers image differ in scale from the others. In addition,  the red channel can be seen. The rest is black. Please verify these charts.

6) Figure 6: what is the value of K1 and K2 and other parameters?

7) The choice of test images is very inconsistent - some are taken for histograms, others for key sensitivity analysis, and still others for the following sections. It would be good to standardize the selection of these images so that each analysis concerns the same ones.

8) Some sentences are not clear, e.g. line 259 "It is a distribution of 3000 pairs as shown in Fig. 7.", line 283-284 "The ideal expectations of and are 99.6094 and 33.463507, respectively [29]." There are more such sentences, so I suggest proofreading.

9) Formula 24: the value should be expressed as a percentage

10) Why such strange values for ideal NPCR and UACI values (99.6094 and 33.463507)? I cannot find confirmation of these values in [29].

11) Table 4: what are the mystery images 4.1.01.tiff etc?

12) Entropy analysis is missing.

13) Comparison of NPCR and UACI with other works is missing.

14) What is the keyspace of the algorithm?

15) Related work can be much more extensive.

Reviewer 5 Report

Dear editor and authors

In this work, the authors propose a family of polynomial chaotic maps that can showcase good chaotic qualities.

The work is indeed very interesting and deals with an important problem. Unfortunately, the authors did not perform an adequate background literature review on existing polynomial chaotic maps. As a result, the map that the propose is almost an exact replica of an existing publication, which the authors do not cite at all

-- Hua, Z., Chen, Y., Bao, H., & Zhou, Y. (2021). Two-dimensional parametric polynomial chaotic system. IEEE Transactions on Systems, Man, and Cybernetics: Systems52(7), 4402-4414.

The present map that is proposed is almost identical to this one, with the only change being one slacar term 'r' in the second equation. The proof is also exactly the same, as the 'r' term does not affect it.

The authors did not cite this work, so they do not suggest that their method is a generalization of that one.

I believe there are other relevant works missing, for example the work 

-- Hua, Z., Zhang, Y., & Zhou, Y. (2020). Two-dimensional modular chaotification system for improving chaos complexity. IEEE Transactions on Signal Processing68, 1937-1949.

is also very relevant and has a similar proof, it should also have been cited.

But it is the first work that unfortunately makes this one much less novel. Unfortunately based on this, I cannot suggest acceptance of this work, regardless of it having good results.

Moreover, here are some additional remarks

--The intro is very good, but it should cite the following work, which discusses exactly these issues in chaos based encryption 

--Teh, J. S., Alawida, M., & Sii, Y. C. (2020). Implementation and practical problems of chaos-based cryptography revisited. Journal of Information Security and Applications50, 102421.

--line 89, typo, it is 'chaotic'

--page 3 there are mistakes in the proof. The Jacobian is wrong, the (1,2) element should have b*gamma, and the (2,2) element should be 'c', not 'b'.

--In (9), what is d_i? The output of the map? But the map has 2 dimensions, which one is it? please specify.

--You mention that you use invertible functions, this is inaccurate. Modulo is not reversible, this is what makes it so effective.

--For table 3, remember that many works apply correlation to a random collection of pixels, so direct comparison is not effective.

--Note that many works use modulo operators as their chaotification method. You can put a discussion on relevant works focusing on modulo operator in the intro.

I would suggest that the authors first study existing works. Focus on poylnomial maps, and also on chaotificationt echniques. Then based on them, they can consider building on the map they already have, to generalize it, so that the novelty is increased, and the technique encompasses previous works as special cases.

If they perform so, i think they can consider resubmitting.

I am attaching the mentioned reference for your convenience. In page 3 you will find the almost identical map.

English is fine, just minor proofreads are required.

Round 2

Reviewer 4 Report

Thanks to the authors for the comprehensive answers. The article has undoubtedly improved in quality. I only have three minor comments:

1) In the 10th comment on the values of npcr and uaci, I wrote about strange optimal values for these measures. It is assumed that the ideal value for npcr is 100, while for uaci it is often 33. Please check it again in several sources.

2) I don't understand why the histograms in Figure 7 for Peepers are black and red. While red is one of the RGB channels, where does black come from? Where are the other channels?

3) By entropy analysis, I meant the Shannon entropy for ciphertexts. As we know, a perfectly encrypted image has an entropy equal to 8 (in practice, the entropy value should be close to 8). It is worth adding such an analysis.

Reviewer 5 Report

Dear authors and editor

In the revised manuscript the authors have made considerable efforts to improve the manuscript. The most important part is that they generalized their map in comparison with the one i pointed out that was almost identical.

Still thoigh the paper has a lot of issues and mistakes, it cannot be accepted as is. The authors must carefully consider my comments, and amend all the errors, otherwise the work should be rejected.

--in the abstract you still mention reversible functions ehich is wrong. You also mention 2d, ehich must be chamged to 3d.

--althought the work by hua et all is mentioned, is it very dismisively provided as just another work. Rhis is very misleading. What you are doing is generalizing their work, so more and direct information sbould be given. It seems like the authors are trying to hide this fact and this is highly unfair.

--line 85 there is a typo.

In the proof there is a mistake in the multiplication of the jacobians. The top right element is not zero of course. Please correct this.

--in section 2.2 obviously rhere should be a diagram of the lyapunov exponents..the whole poont of the paper was that, why make the choice of not including auch a diagram?

--the nist tests include z and y only, why not z as well,? It seems like they left this part unchainged from the previous version which is a mistake since the map has changed.

-the encryption process, you do not mention how are the pixels rearranged and why do you need two sequences to perform that. The reproducibility here by an interested reader is impossible.

-equation 10 there is one parenthesis missing.

-the transmission scheme is not described at all and full of mistakes. In your control scheme, you use actuve control, the most basic method there is. Clearly from formulas 14 this requires knowledge of all three states of the master system. So you need to transmit all these information in the output. This is not the case as you write in 16. Also you do not explain how the synchronisation is possible since one state is tainted by the secret message. Overall very confusing. Reproducibility is impossible.

-line 272 please explain how the key soace is derived.

,,--the differential attach section is completely wrong. These twsts are probably perfoed incorrectly. Your keys are not plaintext dependent and hence such tests will not work. What i guess is that you decide on your own to change the keys each time a new image which one changed pixel is used. This is faulty. The method mist ise plaintext dependent keys for this to work. 

-the complexity analysis is inaccurate. First of all what is n? This is not explained. Also there are two or three processes performed. I am really not sure about the permutation since it is so badly explained. You must analyze the complexity for each step and explain it, not simply write a random and unexplained result.

Overall, very carelessly written sections, but i really like the overall work!. Please be more careful and detailed.

Put weight into reproducibility of your paper .

Some proofreading is required

Round 3

Reviewer 5 Report

In the revised version the authors have succesfully answered all my comments. But one part is still unclear for me.

The control scheme is still unclear

The control process in (14) clearly requires  exact knowledge of x1 and y1 states. So since you transmit y1 and En, knowledge of the exact value of x1 is unavailable at the receiver end. So how is control and synchronziation achieved? This is not properly explained.

Also

--Table 1 there is a typo

--line 206 should be rephrased for clarity
